

# Feasibility assessment of household based small arsenic removal technologies for achieving sustainable development goals

Md Sahadat Hossain[1], Fahima Akhter [2], Victor Emery David Jr.[3]

[1]Department of Thematic Studies-Environmental Change, Linköping University, SE-58183, Linköping, Sweden
[2]School of Economics and Business Administration, Chongqing University, 400030, P.R China
[3]Faculty of Urban Construction and Environmental Engineering, Chongqing University, 400044, P.R China

*Correspondence to*: Md Sahadat Hossain (mdho787@student.liu.se)

**Abstract.** Access to pure drinking water is always occupying as the centric position for long-term sustainable development for all. Although Bangladesh has improved its overall status in drinking water sector compared to 1990 scenario. In 2015, its total safe water sources reached to 87% i.e., still 13% far from full goal achievement. Besides, it has been estimated that 22 of total 164 million population are exposed to >50 to <200 μg/L and 5.6 million are to >200 μg/L respectively. Therefore, achieving sustainable drinking water goals are still challenged for Bangladesh. This study is aimed to assess the hindrance for achieving the entire sustainable development goals (SDGs) and evaluating the role of existing household based Small Arsenic Removal (SAR) technologies in drinking water sector in rural Bangladesh. The literature based evaluation is revealed that SAR technologies have been playing an important role for achieving the SDGs in drinking sectors in Bangladesh. Conversely, the lack of guild lines on their produced sludge laden and consequently improper dumping are adversely affecting the socio-economic and environmental ecosystems. In this vein, there is a framework has been developed based on the relevant studies for achieving the long-term SDGs in the drinking water sector in rural Bangladesh.

**Keywords**: Arsenic, Bangladesh, Removal Technologies, SONO Filter, Sustainability

## 1 Introduction

The global vision for sustainable development has been transformed from Millennium Development Goals (MDGs) into '*Transforming our world: the 2030 Agenda for Sustainable Development*' which will come into force since 1[st] January, 2016. It has adopted as post-2015 development agenda with more comprehensive, sustainability and people-centered goals for solving entire problems (i.e., poverty, inequality, access to water and sanitation etc) for the whole universal. This new universal ambition is consisting 17 goals including 169 targets in which *achieving universal and equitable access to safe and affordable drinking water for all by 2030* (target 6.1) has given precedence as human rights. Moreover, *improving water quality by reducing pollution, eliminating dumping and minimizing release of hazardous chemicals and materials, halving the proportion of untreated wastewater and substantially increasing recycling and safe reuse globally* (Target 6.3) through *supporting and strengthening the local communities in improving water and sanitation management* (Tool 6.b) for sustainable transformation of our mother earth (UN, 2015).



Barkat and Hussam (2008) stated that there are 50% of 150 million people of Bangladesh are in risk of ground water arsenic poisoning. 50% of 30 million households are at arsenicosis risk. Thus, *Action Plan for Poverty Reduction* adopted by Bangladesh governments with the aim of 100% access to drinking water. There are many recent studies on arsenicosis found its chronic and acute effects on human environment in this country. Most of the people are exposed to *melanosis* (dark or

white spot in body); diabetes, high blood pressure, depression, womb and breast-fed child exposure etc in which poor people were found more vulnerable. Women are more exposing than men, suffering from extreme vulnerability due to social discrimination in attending schools, marriage, divorce, family distortion as well as job and working environment resulting for intellectual and economic loss also social imbalance. Consequently, combined initiatives by different groups from their respective platforms have been working for to find the affordable, effective and sustainable solutions.


In this circumstances, the innovation of small household based arsenic filters technology added a significant tool to achieve sustainability in pure drinking water for more than 137 million arsenic vulnerable people of 70 countries all over the world especially in the rural areas. Although its acceptance by different groups and levels (environmentalists, economists, beneficiaries, sociologist, national and international agencies) with optimistic comments for efficiency and effectiveness led

to spread of installation in huge number by stakeholders but have some critic with strong logical arguments after evaluating the long-term sustainability. This study is aimed to assess and evaluate the importance of small household based arsenic filters and necessity of their mass acceptance especially in rural areas of Bangladesh for achieving SDGs. Besides, the end-use fate of these filters and post-treatment residues especially sludge-laden are also taken into in consideration for evaluating the overall sustainability regarding this technology.


The global population's access to improved drinking water had reached into 91% in 2015 in comparison with 76% in 1990. However still almost 1.8 billion people are using faeces contaminated drinking water and 663 million are relying on unimproved sources. Consequently, water-borne diseases i.e., typhoid, dysentery, cholera, diarrhoea and polio in which estimated 50 2000 deaths might be caused by diarrhoea. Moreover, half of the world's population may live in water stressed

areas by 2050. Besides, 38% health care lack of water sources and 35% lack of water in low and middle income countries (WHO, 2015a). Besides, natural and anthropogenic calamities will exacerbate the situation more extremely. Arsenic (As) contamination is one them which has been recognized as the biggest threat for drinking water for developing countries especially Bangladesh (Hassan et al., 2014). Its 59 out of 64 districts are suffering from groundwater arsenic contamination. Consequently, 75 million people are at high risk and 24 million are exposed to this poisoning (Safiuddin and Karim, 2001).

Chakraborti et al. (2015) analyzed the sample of 54,000 tube-wells water and found >50 ug/L in 2200 villages, >10 ug/L in 2700 villages in which 7.5% tube-well were contaminated with >300 ug/L.



## 2 Methodology

This entire study is based on the literature from the indexed journals. The secondary data were retrieved from online database of National Hydro-chemical Survey, 2001; British Geological Survey (BGS) and Department of Public Health Engineering, Government of Bangladesh (DPHE) (DPHE/BGS, 2001) and Multiple Indicator Cluster Survey (MICS) data

conducted by Bangladesh National Drinking Water Quality Survey of 2009 (BNDWQS, 2011) to understand and evaluate the overall arsenic pollution scenario. The relevant articles published in local and national newspapers also considered as the supporting tool for evaluating the severity of arsenic poisoning, necessity, affordability, acceptance and affordability of household filters as well as raising challenges regarding this technology. Finally, a proposed framework is formulated based on scrutinizing the relevant prospects and potentials for achieving SDGs for potable drinking water sector in Bangladesh.

## 3 Necessity of household water treatment technology

The global MDG for halving the proportion of people without sustainable access to potable drinking water sources i.e., ensuring safe drinking water sources for 88% of total population on earth has already been achieved in 2010. In 2015, there are 91% of global population have an improved drinking sources in compared to 76% in 1990. However, there are 663 million people are still lack of improved drinking water sources (WHO, 2015b).

Bangladesh has improved its overall status in drinking water sector compared to 1990 scenario. In 2015, its total safe water sources reached to 87% i.e., still 13% far from full goal achievement (Table 1). The BNDWQS (2011) report estimated that 22 of total 164 million population are exposed to >50 to <200 μg/L and 5.6 million are to >200 μg/L respectively with arsenic poisoning as consuming through their daily drinking water. The preliminary report upon 142.3 million by BNDWQS (2011) found that 19 and 5 million people are suffering from arsenic poisoning for having it >50 to <200 μg/L level in their

drinking water (shown in Table 1). In depth, in rural Bangladesh, 95% of drinking water comes from shallow tube wells, endangered almost 11 million people's (12.6 % of total population) health for having arsenic concentration and looking for hygienic water (Uddin and Huda, 2011). Therefore, challenges for achieving sustainable drinking water goals are still important to be taken into consideration.

The water related infrastructure in Bangladesh is not well developed to govern the system for whole country. The

geographical condition, hard to reach areas and availability of ground water have made its drinking water system more user-centric. It has not only made improved to access for drinking water but also contributing for improved health with providing freedom drinking water for users. But, the emergence of arsenic pollution has challenged overall condition where the rural and poor people are suffering more. As a result, sustainable development goals were also challenged for being prolonged due to arsenic vulnerability. In this consideration, ongoing research and innovation of household based Small Arsenic Removal

(SAR) technologies have created a paradigm for improving the life quality as well as achieving the sustainability in Bangladesh. Being consisting of scientific method and negotiated knowledge, the small household technologies are artefact deployed by experts, efficient, affordable to poor for solving arsenic problem for the socio-economic, environmental and





political interests. The treatment technologies are capable to treat the contaminated water to achieve within the national and WHO standard. A study by Chakraborti et al. (2015) found that contamination is below the national limit (<50 ug/L) and also near about the WHO standard (<10 ug/L) found in deeper tube-wells >200 m. Hence, arsenic pollution mitigation by small household technologies is considered an important tool for social, environmental and economic sustainability in these

rural areas.

## 4 Challenges with SAR Technologies

Arsenicosis was identified in Bangladesh between 1986 to1989. The geo-environmentalist argued its intrusion in the ground water could be effects upstream water withdrawal and/or low discharge of ground water level. They found the inter-linkage between Farakka Barrage (started in 1975) and arsenic poisoning in this region on the basis of time to expose (10 years) of

arsenic and shallow tube-well irrigation channels. The recent scientific eco-toxicological assessment found the significant amount of arsenic concentration in the food chain. Different investigation found the arsenic in rice, vegetables and fruits. The contaminated both organic and inorganic arsenic is intake by human through food affect the ecosystem and human health. According to Joseph et al. (2015), 98% of Bangladeshi people are affecting by arsenic pollution through food and water consumption and 1.15 million people are at cancer risk. The research also revealed that the arsenic contaminated

watering from tube-wells or inland surface water lead to absorb by paddy land, intake by crops and fish species. The bioaccumulation and biomagnifications of arsenic in food chain cause the disruption of ecology of an area.

Undoubtedly, existing available technologies have been helping to achieve the safe drinking water (Hassan et al., 2010). However, these household based Small Arsenic Removal (SAR) Technologies generate significant amount of sludge which disposal or re-treatment processes is still lack of concerns from all respective groups i.e., producers, suppliers, users etc.

(Eriksen and Zinia, 2001). For example, each of the 18 large arsenic and iron treatment plants is yearly generating 170 m$^3$ arsenic laden with the >80-90% treatment capability. In other cases, landfill is the most common technique for these sludge disposal (Basak and Islam, 2008;Mahzuz et al., 2009). The direct disposal leads to contaminate with water and soil releasing toxic elements cause to pollute surface and ground water (Sullivan et al., 2010;Hassan et al., 2014).

## 5. A Case on SONO Arsenic Filter (SAF)

SONO Arsenic Filter (SAF) is a technological determinism which has shaped the vulnerable communities especially in rural areas of Bangladesh. The scientific rhetoric truth on efficiency as well understanding and mass acceptability had led to install 35,0000 filters in Bangladesh, India, Nepal, Egypt and Pakistan for its economic affordability (<0.001 BDTk/L and 1 $ = 77 BDTk approx.) and long time efficiency (>9 years) (Hussam and Munir, 2013). Unfortunately, SAF has obtained very little attention from government's side to incorporate into the national activities along with subsidizing poor people though

deeper attention for mass acceptability has been performing by the invented organization and international organizations. However, it still lack of evaluation information regarding long-term sustainability in SAF's complete life cycle. It is



important to evaluate the long-term sustainability before production, distribution and utilization of this technology. It is true, access to arsenic free pure drinking water through SAF technology has been achieved with the integrated women and political leaders' participation. Conversely, critical analysis of maintenance and sludge management was neglected. Consequently, the questions regarding the long-term future sustainability for open dumping of arsenic coagulated sludge,

risk to be contaminated with biosphere have been arisen. Hence, many environmentalists have argued to promote this arsenic filter because of spreading arsenic far and wide enhanced the destruction of surrounding environment in the warm weather condition of Bangladesh. Some scientists was astonished to find arsenic free tube-well is just 20 feet apart from contaminated one and suggested for using near arsenic free tube-well rather than using a filter. Besides, the affordability is also questioned to buy by poor people with $40-$50 in the affected areas. In these veins, this study developed a framework

(Fig. 1) with considering multi-perspective views in context to socio-economic, environmental, technological, gender dimensions along with feasibility and affordability for using SAR (like SAF) technologies as the important tool for achieving long-term sustainability in drinking sector in Bangladesh.

SAF has been playing unquestionably an important role to combat with arsenic pollution and ensuring pure drinking water in the rural areas of Bangladesh. But recently arose questioned for long-term sustainability has not evaluated yet. In Fig. 2, the

feasibility of SAF has analyzed on the basis of its socio-economic viability, environmentally and technical acceptability. The study found that is socio-economically viable because of low instalment cost ($40-$50), compatible to set-up; local political leaders are concerned for its social importance and also available to buy in the vulnerable areas. This filter does not require any fuel, efficient but produce sludge containing arsenic adsorbed from contaminated water. There are no tools and techniques have not been illustrated for this residual. Technologically it has a satisfactory acceptance due to user

friendliness, longibility, valued local knowledge and technology resulting to be more compatible with the local environment. The SAF users claimed the face difficulty to fix a problem by themselves because little knowledge about it especially women who are the main users of this filter. The most important aspect of this filter is sludge management. The adsorbed arsenic-laden wastes by composite iron matrix (CIM) and coarse sand, fine sand, wood charcoal and bricks are washed or dumped in the open field lead to environmental pollution and degradation.

Therefore, knowledge development might be an important tool for achieving long-term sustainability of this technology. Government and NGO could play an important role in awareness and local capacity building through seminars and workshops. As well, Science and Technological Studies (STS) is important to emphasize the women empowerment and involvement with considering repository of knowledge in relation to socio-economic, cultural, political, scientific and technological perspectives. The understanding the proximate reasons and developing conceptual model might be helpful to

critically addressing the uncertainty and complexity of improper SAF sludge disposal as well as vulnerability assessment through re-examining problems and logical structure for making solutions. Some recent studies have suggested to use long time recycling technique by using these sludge laden might be an option for sludge management e.g., making construction brick by solidification (Hassan et al., 2014;Khan and Yang, 2014;Mahzuz et al., 2009). Besides, the active participation of stakeholders could be ensured for sludge management systems for knowledge and awareness building. In addition, improve



materials in manufacturing process, ensuring more research organizations' involvement, incorporating local community and experts with the government policy making bodies for SAR Technologies might be a unique strategy to produce an effective technological paradigm for future.

## 6 Conclusion

Sustainable development is a complex multi-functional approach of social, economic, environmental, political, cultural, technological and ecological aspects. In this perspective, SAR Technologies have been playing a vital socio-economic and environmental importance in the arsenic vulnerable rural areas. These technologies were derived from the societal need and then shaped that society i.e., it is a social theory incorporating social, economic, environmental, cultural and political aspects. These have ensured pure drinking water to large number of community and protecting exposed women and poor

stakeholders from social discrimination. Arsenic poisoning is a socio-political issue and the long-term sustainability assessment of household based SAR Technologies revealed that its manufacturing, distribution and avoiding its open disposal of arsenic-laden wastes was entirely interconnected and interdependent for achieving SDGs in drinking sector. But the improper management of sludge laden is polluting the surrounding environment and accumulating in the crops and fishes which is also leading to disrupt the ecological system. Although the short-time sustainability in drinking water has been

achieved by the SAR Technologies but the long-term adverse effects on ecosystems made an argument to develop the advanced tools to manage sludge through the participation of different beneficiaries, NGOs, experts and decision making authority. The technological advancement with the combinational local knowledge and skills are needed for long-term sustainability.

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

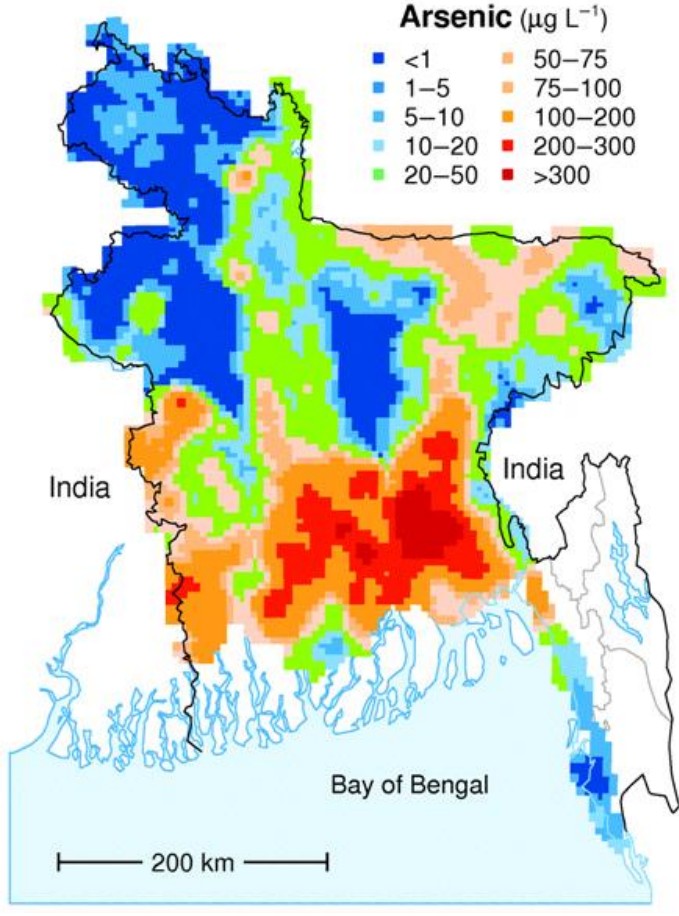

**Figure 1: Distribution of arsenic concentration in drinking water sources in Bangladesh (DPHE/BGS, 2001).**






**Figure 2: Proposed framework for achieving long-term sustainability in SAR Technologies (a case of SAF Technology).**





**Table 1: Access to improved drinking water sources in Bangladesh, 2015 (WHO/UNICEF, 2015)**

| Statistics | Rural | Urban | Total |
|---|---|---|---|
| Total Population (in 1000) | 54984 (34.3%) | 105427 (65.7%) | 160411 (100%) |
| Safe drinking coverage (in 1000) | 47836 (87%) | 91722 (87%) | 139558 (87%) |
| Lack of access to safe drinking water sources (in 1000) | 7148 (13%) | 13706 (13%) | 20853 (13%) |

**Table 2. Access to improved drinking water sources in Bangladesh, 2015**

| Arsenic concentration (μg/L) | (DPHE/BGS, 2001) exposure proportion (Tested samples, 3534) | (BNDWQS, 2011) exposure proportion (Tested samples, 3534) | Exposed Population (BNDWQS, 2011) (estimated on 142.3 million) | Exposed Population by (BNDWQS, 2011) (estimated on 164 million) |
|---|---|---|---|---|
| 0–10 | 57.9 | 68 | 82.3917 | 94.956 |
| 10.1–50 | 17.1 | 18.7 | 24.3333 | 28.044 |
| 50.1–100 | 8.9 | 7.2 | 12.6647 | 14.596 |
| 150.1–200 | 4.2 | 1.4 | 5.9766 | 6.888 |
| 200.1–250 | 2.9 | 1.4 | 4.1267 | 4.756 |
| 250.1–300 | 2.1 | 1.1 | 2.9883 | 3.444 |
| 300+ | 1.8 | 0.4 | 2.5614 | 2.952 |