# Peer review of "Feasibility assessment of household based small arsenic removal technologies for achieving sustainable development goals"

_Drinking Water Engineering and Science, 2016_

## Referee Comment (RC1) · A. Ghosh (Referee) · 27 Jun 2016

1.After section 4.1 there is no section 4.2; please consider changing the section numbering. 2.Mechanism for operation and maintenance of SAF should be elaborated. 3. Why Fig 1 has been referred in the following paragraph? :

"In these veins, this study developed a framework (Fig. 1) with considering multi-perspective views in context to socio-economic, environmental, technological, gender dimensions along with feasibility and affordability for using SAR (like SAF) technologies as the important tool for"

---

## Referee Comment (RC2) · Anonymous Referee #2 · 12 Jul 2016

General comments The article addresses an interesting and urgent societal challenge related to drinking water supply, and is therefore closely linking to the scope of the journal. However, before it may be accepted for publication it is crucial that the authors revise the article as such that it may qualify as a scientific review. At present, the publication cites only 5 peer-reviewed publications, which is - in my opinion - insufficient to classify as a scientific review article. I would encourage the authors to restructure their article in such a way that it includes more citations in the field of arsenic removal in Bangladesh. Perhaps by searching for terms related to "household water treatment and storage", read-f, alcan, sidko, etc. the authors can extend their review further. There have been multiple publications in journals like Water Resaerch, ES&T and Water Science and Technology that should be added to the review. After extending the review, I would encourage the authors to re-submit the manuscript.

Specific comments related to the abstract: l8: consider changing "pure" to "safe" l15: "important role" - based on what information was determined that there was indeed an important role for SAR. I could not find this in the manuscript. l16 giuld lines = guidelines l17 "there is" seems misplaced

---

## Author Comment (AC1) · 28 Jul 2016

We are thankful for the generous suggestion. We strongly believe that revision based on your constructive suggestion will help to enhance the quality of our manuscript.

Comment-1. After section 4.1 there is no section 4.2; please consider changing the section numbering.

Response: We are agreed with this change.

2.The mechanism for operation and maintenance of SAF should be elaborated.

Response: Agreed.

[Figure]

3. Why Fig 1 has been referred in the following paragraph? : "In these veins, this study developed a framework (Fig. 1) with considering multi-perspective views in context to socio-economic, environmental, technological, gender dimensions along with feasibility and affordability of using SAR (like SAF) technologies as the important tool for"

Response: It was our unanticipated mistake and apologizing for it. Now we have revised and placing (Fig 1) in the appropriate referable paragraph.

---

## Author Comment (AC2) · 28 Jul 2016

Response Letter to Referee 2

We are thankful for generous comments. We have revised our manuscript according to your generous comments and suggestions.

General comments:

The article addresses an interesting and urgent societal challenge related to drinking water supply, and is therefore closely linking to the scope of the journal. However, before it may be accepted for publication it is crucial that the authors revise the article as such that it may qualify as a scientific review.

Response: Agreed and thank you for the truthful reply.

Comment: At present, the publication cites only 5 peer-reviewed publications, which is - in my opinion – insufficient to classify as a scientific review article.

Response: Agreed.

Comment: I would encourage the authors to restructure their article in such a way that it includes more citations in the field of arsenic removal in Bangladesh. Perhaps by searching for terms related to "household water treatment and storage", read-f, alcan, sidko, etc. the authors can extend their review further.

Response: Agreed and thankful for your generous suggestion.

Comment: There have been multiple publications in journals like Water Research, ES&T and Water Science and Technology that should be added to the review. After extending the review, I would encourage the authors to re-submit the manuscript.

Response: We are agreed with your proposal and have started for enhancing our manuscript according to your comments and suggestions.

Specific comments related to the abstract:

Comment: l8: consider changing "pure" to "safe"

Response: Agreed.

l15: "important role" - based on what information was determined that there was indeed an important role for SAR. I could not find this in the manuscript.

Response: SAR has been playing an important role to the 'rural arsenic affected poor people' as an option for accessing to arsenic-free drinking water.

We are revising our manuscript to make the arguments more clear to the readers.

Comment: l16 giuld lines = guidelines

Response: Revised.

Comment: l17 "there is" seems misplaced

Response: Revised.

―――――――――――――――――――――――